# *Salmo salar* Skin and Gill Microbiome during *Piscirickettsia salmonis* Infection

**DOI:** 10.3390/ani14010097

**Published:** 2023-12-27

**Authors:** Marcos Godoy, Yoandy Coca, Rudy Suárez, Marco Montes de Oca, Jacob W. Bledsoe, Ian Burbulis, Diego Caro, Juan Pablo Pontigo, Vinicius Maracaja-Coutinho, Raúl Arias-Carrasco, Leonardo Rodríguez-Córdova, César Sáez-Navarrete

**Affiliations:** 1Centro de Investigaciones Biológicas Aplicadas (CIBA), Lago Panguipulli 1390, Puerto Montt 5480000, Región de Los Lagos, Chile; marco.montesdeoca@ciba.cl (M.M.d.O.); diego.caro@ciba.cl (D.C.); 2Laboratorio de Biotecnología, Facultad de Ciencias de la Naturaleza, Escuela de Medicina Veterinaria, Universidad San Sebastián, Sede Patagonia, Lago Panguipulli 1390, Puerto Montt 5480000, Región de Los Lagos, Chile; 3Doctorado en Ciencias de la Ingeniería, Departamento de Ingeniería Química y Bioprocesos, Escuela de Ingeniería, Pontificia Universidad Católica de Chile, Avenida Vicuña Mackenna 4860, Santiago 7820436, Macul, Chile; ycoca@uc.cl; 4Programa de Magíster en Acuicultura, Facultad de Ciencias del Mar, Universidad Católica del Norte, Coquimbo 1780000, Elqui, Chile; rudysuarez.vet@gmail.com; 5Department of Animal, Veterinary, and Food Sciences, Aquaculture Research Institute, University of Idaho, Hagerman, ID 83332, USA; bledsoe@uidaho.edu; 6Facultad de Medicina y Ciencia, Centro de Investigación Biomédica, Universidad San Sebastián, Sede Patagonia, Lago Panguipulli 1390, Puerto Montt 5480000, Región de Los Lagos, Chile; ian.burbulis@uss.cl; 7Laboratorio Institucional, Facultad de Ciencias de la Naturaleza, Escuela de Medicina Veterinaria, Universidad San Sebastián, Sede Patagonia, Lago Panguipulli 1390, Puerto Montt 5480000, Región de Los Lagos, Chile; juan.pontigo@uss.cl; 8Unidad de Genómica Avanzada, Facultad de Ciencias Químicas y Farmacéuticas, Universidad de Chile, Santiago 7820436, Macul, Chile; vinicius.maracaja@uchile.cl; 9Centro de Modelamiento Molecular, Biofísica y Bioinformática (CM2B2), Facultad de Ciencias Químicas y Farmacéuticas, Universidad de Chile, Santiago 7820436, Macul, Chile; 10Beagle Bioinformatics, Santiago 7820436, Macul, Chile; 11Programa Institucional de Fomento a la Investigación, Desarrollo e Innovación (PIDi), Universidad Tecnológica Metropolitana, Santiago 7820436, Macul, Chile; raul.arias@utem.cl; 12Escuela de Ingeniería, Facultad de Ingeniería, Universidad Santo Tomas, Santiago 7820436, Macul, Chile; lrodriguez11@santotomas.cl; 13Departamento de Ingeniería Química y Bioprocesos, Escuela de Ingeniería, Pontificia Universidad Católica de Chile, Av. Vicuña Mackenna 4860, Santiago 7820436, Macul, Chile; csaez@ing.puc.cl; 14Centro de Investigación en Nanotecnología y Materiales Avanzados (CIEN-UC), Pontificia Universidad Católica de Chile, Av. Vicuña Mackenna 4860, Santiago 7820436, Macul, Chile

**Keywords:** *Piscirickettsia salmonis*, *Salmo salar*, skin, ulcer, microbiome

## Abstract

**Simple Summary:**

Farmed Atlantic salmon are routinely exposed to bacterial pathogens, e.g., *Piscirickettsia salmonis*. Infection by *Piscirickettsia* sp. leads to a complex array of skin ulcers that can be difficult to treat and increases susceptibility to opportunistic infections. Evidence indicates that bacterial networks residing on salmon skin protect against developing ulcers by excluding pathogen colonization. A collapse of these beneficial interactions is thought to promote susceptibility to pathogen colonization during early stages of infection. We characterized the types and abundances of bacterial constituents on the skin of healthy Atlantic salmon compared with fish suffering from *P. salmonis* infection to test this hypothesis. The knowledge we gained can be used to optimize methods for early detection and prevention of skin ulcers by disrupting cooperative interactions between pathogenic bacteria.

**Abstract:**

Maintaining the high overall health of farmed animals is a central tenant of their well-being and care. Intense animal crowding in aquaculture promotes animal morbidity especially in the absence of straightforward methods for monitoring their health. Here, we used bacterial 16S ribosomal RNA gene sequencing to measure bacterial population dynamics during *P. salmonis* infection. We observed a complex bacterial community consisting of a previously undescribed core pathobiome. Notably, we detected *Aliivibrio wodanis* and *Tenacibaculum dicentrarchi* on the skin ulcers of salmon infected with *P. salmonis*, while *Vibrio* spp. were enriched on infected gills. The prevalence of these co-occurring networks indicated that coinfection with other pathogens may enhance *P. salmonis* pathogenicity.

## 1. Introduction

Salmonid rickettsial septicemia (SRS) is an intracellular infection caused by *P. salmonis*. This disease causes great morbidity and mortality among salmonid species, especially Atlantic salmon (*Salmo salar*) [1], and other fish species globally that include *Epinephelus melanostigma* [2], *Atractoscion nobilis* [3], and *Dicentrarchus labrax* [4]. Manifestations of SRS include ulcerative, systemic, and granulomatous presentations that include muscle necrosis [5,6,7]. Clinical SRS is a leading cause of both fish mortality and biomass loss in the salmon industry, which promotes the excessive use of antibiotics. Pathogenic bacteria from different families are thought to interact in a manner that promotes virulence, but the composition and enrichment of co-occurring bacterial species on the skin and mucosal layers of *P. salmonis*-infected fish is not known. Reducing fish morbidity and increasing health and wellbeing is a clear goal of the aquaculture field, but the lack of knowledge about resident bacterial communities on healthy and *P. salmonis*-infected fish skin prevents achievement of this goal.

Skin disorders are a prominent issue in aquaculture. Monitoring fish skin for ulcers is essential for maintaining fish health. Changes in skin surfaces, including ulcers, are important indicators of underlying health problems. Approximately 1.1–2.5% of farmed fish succumb to ulceration [8] and an additional 0.7–3.8% are down-classified at harvest due to reduced market quality [9].

Fish skin serves as a protective and dynamic tissue layer in constant interaction with the environment. It houses host-derived antimicrobial compounds and immunological components, acting as the primary defense against infectious pathogens [10,11]. Salmon skin is covered by a secreting mucosa. This layer is colonized by a complex microbial community. Evidence indicates that positive associations between mucosa-associated bacteria and the host immune system protects the fish from colonization by pathogenic bacteria [12]. Shifts in the skin microbiota due to environmental changes can lead to the dominance of opportunistic pathogenic bacteria, posing a threat to fish health [13]. The fish skin microbiome, in constant contact with the aquatic environment, facilitates microbial exchange with surrounding water and potentially other hosts [14]. Fish gills have a significant surface area per unit weight and represent the largest organ interfacing with the environment in teleost fish [15,16]. This feature makes them useful for monitoring fish health [17]. Some researchers such as Koppang et al., 2015 and Streit, 1998 [15,16], estimated that fish gills represent around 0.1 to 0.4 m^2^/kg of body weight. Gill tissues react quickly to unfavorable environmental conditions [17]. Several factors influence the composition of the gill microbiome, such as the fish health status, nutritional intake and condition of the water [18,19,20]. Information on the gill microbiome is scarce, and comprehensive comparisons across various fish species, focusing on the gill and its external environment, are not widely available [21]. The rapid and acute responsiveness of gills to external alterations positions them as an optimal organ for monitoring fish health [22] and a possible point for identifying microbial biomarkers associated with gill health. The same also applies to fish skin, due to its high sensitivity. Understanding the dynamics of microbial communities within both the skin and gills is instrumental not only for the management of wild fish populations but also for enhancing the performance and growth of captive fish. This can facilitate the swift diagnosis of fish diseases, ultimately contributing to more effective aquaculture practices [18,23].

Human skin microbiome [24,25,26] and fish gut microbiome have been extensively studied while investigations of skin and gill microbiomes in salmonids remain limited [27]. The body of literature focusing on the *Salmo salar* skin microbiome and its dynamics during infection is notably sparse. Some studies have delved into gill microbiomes, exploring aspects such as the influence of *Flavobacterium psychrophilum* on susceptible and resistant lines [28] and the effects of ploidy on salmonid alphavirus infection [29]. For instance, Casadei et al., 2023 [30] conducted an evaluation of microbiota changes in groups of Atlantic salmon (*Salmo salar*), including both females and males, following treatment to eliminate infections caused by the ectoparasite *Lepeophtheirus salmonis*, commonly known as sea lice. In a study conducted in Chile [31], Valenzuela-Miranda et al., 2022 investigated the sea lice *Caligus rogercresseyi* as a possible vector for pathogenic bacterial groups within *S. salar* farms. Despite the evident significance of these mucosal microbiomes, there remains a relative dearth of research focusing on the microbiomes of salmonid skin and gills. This knowledge gap underscores the need for further investigations to unravel the complexities of these microbiomes and their responses to various environmental stressors and infectious agents.

For many years, it has been well recognized that microorganisms have a higher chance of survival within communities. However, this concept has long been overlooked in fish health diagnosis. The prevailing paradigm in aquaculture, influenced by Koch’s postulate [32], has traditionally focused on a one-pathogen, one-disease framework. Consequently, the diagnosis of coinfections involving fungal, bacterial, or viral agents in fish farms has been scarce. There are several factors contributing to this oversight, including the lack of adequate analytical techniques capable of detecting multiple pathogens simultaneously, heavy reliance on traditional diagnostic methods primarily based on histological and pathological criteria, suboptimal sampling procedures, inadequate sample transportation, and a prevailing focus on analyzing fish farm mortality data. Some experts have emphasized the potential threats posed to aquatic animals by coinfections involving opportunistic pathogens, which can often persist and spread in aquatic environments, exacerbating health problems in wider fish populations [33]. In the context of such coinfections, *P. salmonis* stands as an illustrative case, with reports of coinfections involving this pathogen being notably scarce to date. Limited investigations have explored the relationship between *P. salmonis* and the parasite *C. rogercresseyi* [34,35,36]. Another study delved into the presence of different genogroups of the *P. salmonis* pathogen in Atlantic salmon samples collected from farms that experienced outbreaks of two distinct *P. salmonis* genogroups [37].

Knowledge of the composition and abundance of bacterial communities on fish skin and gills is crucial for improving fish health. Whether *P. salmonis* cooperates with other bacteria to promote virulence and infection is not known. However, cooperativity between pathogenic bacteria can be established by measuring the enrichment and co-occurrence of different bacterial species in states of disease, like skin ulcers. Here, our goal was to characterize the microbiome of healthy Atlantic salmon skin/gills and monitor how the structure of bacterial networks change during *P. salmonis* infection.

## 2. Materials and Methods

### 2.1. Fish Sampling

We conducted our analysis of Atlantic salmon (*S. salar*) during the grow-out phase, the average weight of the sampled population was between 2.8 kg and 3.1 kg. Our focus was on ulcerated salmon, selected from seven distinct Chilean salmon farming sites denoted as C1, C2, C3, C4, C5, C6, and C7, all located in the Los Lagos and Aysén Regions (Table 1). We sampled a total of 42 individuals of Atlantic salmon (*S. salar*), all of which were identified as infected across these farming locations. To establish a control group, we randomly chose twelve healthy fish from farm sites C8 and C9.

Our field sampling involved collecting three types of samples for microbiome analysis. Initially, we gathered swabs from Atlantic salmon (*S. salar*) individuals displaying skin ulcerations, swabs from the non-ulcerated skin areas of those fish with ulcers, and swabs from the first-gill arch of each fish. In the control group, we took samples from the midline of the fish’s skin and their first-gill arch. These samples were carefully stored in RNAlater transport medium within sterile Eppendorf tubes.

Samples of the anterior kidney, heart, and spleen were obtained from the salmon at farms (C1 to C7) to confirm the presence of *P. salmonis* in the fish from the infected centers. This confirmation was achieved through PCR analysis, following the method outlined by Karatas et al., 2008 [38]. Samples from farms C8 and C9 were also collected to validate the negative status of the control groups.

Finally, we collected samples of both skin and ulcerated skin for histological analysis. These tissue samples were preserved in 10% buffered formalin and were processed according to standard procedures. Sections measuring 3–4 mm were prepared and stained with hematoxylin and eosin (H&E), as per the method outlined by Prophet et al., 1992 [39]. This process allowed us to describe significant microscopic morphological changes.

### 2.2. Gross Pathology and Immunohistochemistry for Detecting Piscirickettsia salmonis in Skin Tissue

Veterinarians from the Centro de Investigaciones Biológicas Aplicadas (CIBA) carried out the gross pathology analysis, documenting external and internal gross lesions. Frequency of occurrence was determined using the methods outlined by Noga, 2010 [40]. Sections of healthy skin and ulcerated skin were preserved in 10% buffered formalin and subjected to immunohistochemistry staining to detect *P. salmonis* antigens. In our study, the detection of *P. salmonis* was achieved through immunohistochemical techniques adapted from the guidelines presented by Prophet et al. in 1992 [39]. We initiated the process by utilizing the monoclonal anti-*Piscirickettsia salmonis* antibody (Clone 7G4/D9) diluted 1:500 to specifically target the pathogen within skin tissue sections. These sections, sliced to a thickness of 3 μm from paraffin-embedded samples, were prepared on slides and left to dry overnight.

The deparaffinization step involved three successive xylene washes, each lasting three minutes. We then transitioned to a series of ethanol solutions, decreasing in concentration from 100% to 50%, to rehydrate the sections. This was followed by rinsing in deionized water. For the antigen retrieval phase, we placed the slides in a steaming container with sodium citrate buffer for 40 min. After a quick rinse in phosphate buffered saline (PBS) provided by Merck (Santiago de Chile, Chile), we applied a 3% hydrogen peroxide solution for 10 min within a humid chamber to block endogenous peroxidase activity. After a subsequent rinse through immersion in PBS, we blocked endogenous proteins using either a protein blocker or bovine serum albumin (BSA protein) sourced from Thermo Fisher, also for a duration of 10 min, in a moist environment. The primary antibody incubation was conducted with the Clone 7G4/D9 monoclonal antibody at a 1:500 dilution for 40 min, maintaining the slides in a humid chamber heated to 45 °C to prevent drying. Post-incubation, we utilized the HiDef Detection™ HRP Polymer System (Sigma Merck, St. Louis, MO, USA, Product Number: 954D), following the manufacturer’s guidelines. This step was crucial for amplifying the visualization of the antigen–antibody interactions within the tissue sections.

Subsequent to this, we halted the staining process by washing the slides with distilled water. Counterstaining was performed using Mayer’s hematoxylin for three minutes, followed by rinsing in tap water for five minutes. The final preparation for microscopic examination involved mounting the slides with a glycerol-based, water-soluble mounting solution. Control slides, which included *P. salmonis* tissues, were processed identically, except for the substitution of the primary antibody with normal rabbit serum.

### 2.3. DNA Isolation, 16S rRNA Gene Amplification, and Sequencing

DNA was isolated from tissue samples of skin, ulcers, and gills using the Qiagen DNA Microbiome extraction kit (Qiagen, Catalog No. [51704]), following the guidelines provided by the kit manufacturer. DNA concentration was determined using 2 μL of each sample, utilizing the Invitrogen Qubit dsDNA BR Assay kit (Invitrogen, Catalog No. [Q33265]). We amplified the V4 hypervariable region of the bacterial 16S rRNA gene from each DNA sample. This was achieved using a primer set specifically targeting the V4 region: 515F/806R (5′-GTGCCAGCMGCCGCGGTAA-3′/5′-GGTACHVGGGTWTCTAAT-3′). The PCR process for each sample involved a 35-cycle reaction using the HotStarTaq Plus Master Mix Kit (Qiagen, Catalog No. [203645]), with a specific temperature and time protocol. Following PCR, the amplicons from different samples were combined in equal measures and purified. Sequencing was then performed on the Illumina NovaSeq system, adhering to the provided kit protocols. Finally, all the sequenced data were uploaded to the NCBI Sequence Read Archive under the BioProject code PRJNA1044012.

### 2.4. Data Filtering, Amplicon Sequence Variants Production, and Taxonomic Assignment

The raw sequencing data were processed and refined using Fastp version 0.20.1 [41], which involved trimming the initial 10 nucleotides and excluding sequences containing any ambiguous nucleotides (Ns). A sliding window approach from right to left, with a window size of 4, was employed to eliminate fragments with a quality score (Q-score) below 30. The resulting high-quality sequences were then subjected to de-noising, removal of chimeric PCR artifacts, and merging. This was accomplished using DADA2 version 2.1.18 [42], a package available in R, resulting in refined 16S amplicon sequence variants (ASVs). Taxonomic classification of these ASVs was carried out using the same R package, DADA2, with the SILVA database version 138.1 [43] as the reference. The classified taxonomic data and ASV read counts were compiled into a phyloseq version 1.4 [44] object to facilitate subsequent analyses. For assessing alpha diversity, we applied rarefaction to the samples and analyzed them using the phyloseq version 1.4 package in R. To visualize the diversity, rarefaction curves were generated using the “rarecurve” function from the vegan package version 2.6.2 [45] in R.

### 2.5. Beta Diversity: Nonmetric Multidimensional Scaling (NMDS) Analysis

To analyze 16S rRNA amplicon sequencing variants (ASVs), we employed non-metric multidimensional scaling (NMDS) using the Bray–Curtis method. The Bray–Curtis dissimilarity index was computed for each sample using the “vegan” R package, specifically employing the “adonis2” function. Subsequent NMDS plots displayed differentiation between controls and fish with skin ulcers. The skin, gills, and ulcers of infected fish were represented as green dots, while the skin and gill mucosa of control salmon were shown as blue-green dots. Furthermore, permutational multivariate analysis of variance (PERMANOVA) was conducted on distance matrices (*p* = 0.001), underscoring dissimilarities (beta diversity) between the two groups under investigation: the ‘healthy’ Atlantic salmon control group and the *P. salmonis*-infected Atlantic salmon.

### 2.6. Alpha Diversity: Shannon Index

To assess species diversity within the ecological communities present in our samples, we employed the Shannon diversity index, also known as the Shannon–Wiener index. This metric elucidates two principal facets of diversity: species richness and the even distribution of individuals across these species. Calculations were executed using the “vegan” R package across the five investigated tissues. Subsequently, Bray–Curtis analysis was performed at the tissue level, comparing the “healthy” Atlantic salmon control group to the *P. salmonis*-infected Atlantic salmon group (*p* = 0.001). Notably, the resulting analysis presented distinct separations among the sample types, even though some overlap was observed between them.

### 2.7. Linear Discriminant Analysis (LDA)

We utilized the LEfSe (linear discriminant analysis effect size) algorithm [46] to discern significant taxonomic differences among tissues. The LEfSe algorithm involves performing a Wilcoxon test for each taxonomic group and identifying amplicon sequence variants (ASVs) that stand out in their abundance, which, in line with the criteria set for our study, consisted of ASVs with *p*-values less than 0.05. Employing this approach enabled us to pinpoint ASVs that played a key role in differentiating the microbial profiles of the respective tissues.

### 2.8. Co-Occurrence Network Analyses

We employed cooccur v1.3 [47] for in-depth co-occurrence network analysis ensuring adherence to a stringent *p*-value threshold of 0.05. The derived networks were then rendered visible with visNetwork v2.1 (https://datastorm-open.github.io/visNetwork/) accessed in 2021. Further examination and insights into these networks were achieved through the NetShift v1 [48] tool. A probabilistic model was utilized in our co-occurrence network analysis to further delve into the inter-relationships of operational taxonomic units (ASVs) within the healthy skin and ulcers of Atlantic salmon. Comparative networks for infected gills vs. healthy tissue and infected skin vs. healthy tissue were also studied, with the results presented in Appendix Figure A4 and Figure A5. The aim in this stage of our analysis was to assess the prevalence and patterns of ASV co-occurrences within each tissue type. These co-occurrences were categorized as follows:

Positive Associations (Green): Signifying co-occurrence frequencies that surpassed expectations, hinting at potential cooperative interactions between ASVs.

Negative Associations (Red): Indicating co-occurrence frequencies below expected levels, suggesting potential competitive or exclusionary relationships.

Random Associations (Blue): Representing co-occurrence frequencies that aligned with expectations.

Within the network structure, the size of each node was determined based on the scaled NESH (neighbor shift) score. Nodes that shift significantly from the control to the case scenario, denoted by a change in color to red, are identified as “key drivers” or “key nodes”. These key nodes play a pivotal role in modulating the structure of microbial interactions within the network. Conversely, nodes marked with black dots represent taxa that exhibit less significant shifts in their local neighborhoods, indicating a relatively lower influence on the network’s bacterial dynamics. 

## 3. Results

### 3.1. Gross Pathology and Immunohistochemistry, PCR

The fish exhibited scaling, raised scales, and single to coalescent ulcers with a distinct white border, exposure of musculature, and variable size (Figure 1). Internally, the fish showed an absence of food in the digestive system, splenomegaly, and varying degrees of renomegaly. Visceral fat congestion was observed, and in a reduced percentage of the sampled fish, hepatomegaly and white nodules were present in the liver. Histologically, the analyzed cutaneous tissue showed a disruption of the epidermis and partial involvement of the dermis. Additionally, inflammation, hemorrhages, and necrosis of the epidermis, dermis, and hypodermis were observed. Application of the immunohistochemical (IHC) technique using monoclonal antibodies specific for *P. salmonis* on paraffin-embedded tissues revealed a positive reaction (brown color), indicating the presence of coccoid structures consistent with the presence of *P. salmonis* in the skin lesions (Figure 2).

### 3.2. Beta Diversity

#### Bray–Curtis Method

Figure 3 shows a non-metric multidimensional scaling (NMDS) plot derived from Bray–Curtis analysis, delineating the microbial communities of Atlantic salmon under two distinct conditions: healthy and *P. salmonis*-infected. The microbiota profile of healthy salmon, represented by red dots, show predominant occupation in the lower-right quadrant of the plot, with a relatively tight clustering suggesting a consistent microbial community composition among the healthy samples. Conversely, the microbiota associated with *P. salmonis*-infected salmon, depicted as blue-green dots, are largely situated in the center left quadrants. This distribution might indicate varying degrees of infection among the samples, as suggested by the more dispersed nature of the blue-green cluster. The notable intersection of red and blue-green dots in the central region of the plot underscores the microbial communities common under both conditions, potentially hinting at a transitional microbiota state or the presence of foundational microbes resilient to infection. The axes NMDS1 (*X*-axis) and NMDS2 (*Y*-axis), spanning values from −0.50 to 0.50, facilitate the spatial representation of microbial dissimilarity. In the NMDS plot, the spatial arrangement of points carries significance. Points situated closely together signify greater similarity in microbial community composition, while those spaced further apart suggest distinct microbial compositions. Figure 3 offers an in-depth juxtaposition of the microbial landscapes in the salmon under the two conditions.

In Figure 4, the Nonmetric Multidimensional Scaling (NMDS) plot derived from the Bray–Curtis dissimilarity matrix visualizes microbial community diversities across various tissue levels in Atlantic salmon. For each data point, its placement on the NMDS plot represents the microbial composition, with the position on the NMDS1 (horizontal) and NMDS2 (vertical) axes indicating the primary axes of microbial variation.

Healthy skin samples, depicted in olive green, are distributed primarily in the right quadrant; yet they also extend towards the lower-right section, indicating a wider range of variation on the NMDS2 axis. This variation suggests a broader microbial composition within healthy skin tissues. Conversely, the infected skin samples, shown in blue, are dispersed across the plot, with a notable concentration on the left side. This indicates a significant presence of these communities within the negative range of the NMDS1 axis. Within this category, a subgroup located in the lower-left quadrant is particularly distinct. The proximal positioning of these Infected skin samples and healthy skin samples on the plot suggests shared microbial taxa but with evident compositional nuances.

The Infected ulcer samples, represented in purple, are positioned toward the lower values on the NMDS2 axis. Their distribution spans positive and negative values on the NMDS2 axis, underscoring a diverse range of microbial communities associated with ulcerated conditions. The significant departure of their microbial communities from those of healthy tissues may indicate a unique microbial consortium associated with ulcerated conditions.

A clear divergence is observed between healthy gills colored in salmon pink and infected gill samples represented in green, along the NMDS1 axis. Healthy gill samples preferentially occupy the upper quadrant but are not exclusively confined to the upper-right. They are scattered throughout the upper half of the plot, demonstrating positive values on both NMDS axes. In stark contrast, the infected gill samples are more centrally located, with many extending into the negative domain of the NMDS1 axis. This spatial demarcation reflects the microbial perturbations induced by *P. salmonis* infections in gill tissues.

The schematic representation of the samples in Figure 4b elucidates the underlying microbial dynamics across various tissue types in Atlantic salmon. The discrete placement of each sample along the Non-metric Multidimensional Scaling (NMDS) axes underscores both the diversity and the parallels of microbial assemblies. 

### 3.3. Alpha Diversity

#### Shannon Index

Following alpha diversity analysis of the microbial profiles comparing healthy skin (indicated in olive green) and infected ulcers (illustrated in purple), a pronounced differentiation was observed (Figure 5). A Wilcoxon test corroborated this observation, showing a statistically significant divergence with a *p*-value of 0.00014. This divergence suggests a profound alteration in the microbial ecosystem of the skin post-ulceration, likely indicative of a shift in the dominant microbial species or the emergence of rare taxa.

Contrastingly, when examining the microbial landscape of healthy skin (olive green) against infected skin without ulceration (blue), the divergence, though less pronounced, remains significant. A *p*-value of 0.042 for the Wilcoxon test reaffirmed this distinction. It hints at initial changes in the microbial composition due to infection, even before ulcerative manifestations appear.

The results from comparison of the microbial profiles between infected skin (represented in blue) and infected ulcers (depicted in purple) introduce additional intricacy. Although both conditions result from infection, they display distinct microbial diversities, as underscored by a *p*-value of 0.0016. Such variance indicates the microbiota’s dynamic adaptation or response to advancing stages of the disease.

Lastly, upon inspection of the healthy gills (depicted in salmon pink) and the *P. salmonis*-infected gills (shown in green), a distinct pattern emerges. Both gill tissues appear to have similar alpha diversity. However, the alpha diversity values revealed that the gills of infected fish possessed a numerically higher level of diversity. The underlying reasons for this increased diversity are not immediately apparent, but it could indicate a microbial defense mechanism in action, or perhaps it is the result of specific microbial species taking advantage of the infected environment.

### 3.4. Taxonomic Assignment

Figure 6 presents the relative abundance of the bacterial microbiome. At the phylum level, each analyzed tissue in Atlantic salmon exhibits a prevalence of the phylum Proteobacteria (74.41%), followed by the phyla Bacteroidota (5.33%), Verrucomicrobiota (3.59%), and Campylobacterota (2.61%). Furthermore, at the order level, a clear differentiation is evident among the samples analyzed. Notably, the presence of the Piscirickettsiales, Flavobacteriales, and Pseudomonales orders was observed in the infected skin and ulcer samples. Furthermore, an overabundance of the genus *Pseudomonas* spp. was noted in the infected microbiome samples. These particular bacterial groups are commonly associated with freshwater infections and are often indicative of microbial dysbiosis in fish.

At the species level, our primary aim was to detect the presence of the pathogen *P. salmonis* within each of the analyzed tissues. To achieve this, we analyzed the amplicon sequence variants (ASVs) specific to this microorganism, as illustrated in Figure 7. Our examination revealed the presence of *P. salmonis* in all three infected tissues, with ASV counts of 48 in the gills, 54 in the skin, and 53 in the ulcers. Importantly, no ASVs (N.D) corresponding to the pathogen were detected in the tissues of the control group. This same analysis was extended to the level of bacterial families for all tissues, as presented in Appendix Figure A3, further confirming this observation.

### 3.5. Linear Discriminant Analysis (LDA)

From the LDA analysis, it is evident that the ulcers in the Atlantic salmon were primarily colonized by *P. salmonis*, *Tenacibaculum dicentrarchi*, and *Aliivibrio wodanis* (Figure 8). This particular grouping of bacteria indicates a close correlation between these taxa and the ulcerative state observed in Atlantic salmon. In the infected skin tissue, there is a significant presence of genus *Vibrio* spp. The predominance of this genus in infected skin tissue suggests its potential role in skin infections in salmon. On the other hand, healthy skin tissue showcases a more diverse microbial profile, characterized by taxa such as Photobacterium, Psychrobacter, and Ralstonia. This assortment of microbes implies a balanced microbial community, possibly contributing to the health and protective functions of the salmon’s skin. Observing this genus in both infected skin and gill tissues might indicate a shared pathogenic route or mechanism affecting both these sites during an infection episode. Contrarily, the microbial profile of the healthy gill, as inferred from the LDA analysis, is markedly distinct from its infected counterpart. The noticeable lack of explicitly pathogenic taxa like *Vibrio* spp. in healthy gills may indicate equilibrium in terms of microbial composition.

### 3.6. Co-Occurrence Network

Our analysis of microbial co-occurrence networks in Atlantic salmon revealed intricate patterns. For healthy skin (Figure 9), we discerned a substantial network comprising 3051 positive and 161 negative interactions, with the dominance of positive correlations hinting at pronounced microbial diversity-dependent effects in maintaining skin health. These findings are consistent with Friedman and Alm, 2012 [49], indicating that two perfectly correlated ASVs are operating in synergy. Within this framework, families like Rhodocyclaceae, Micrococcaceae, and Carnobacteriaceae, marked by high NESH scores, stand out as potential beneficial microbes for skin health.

In our analysis of the gills (Appendix Figure A4) characterized by a relatively lower bacterial abundance, 161 positive and 87 negative interactions were identified. Taxa with significant NESH scores—specifically Micrococcaceae, Rhizobiaceae, and Akkermansiaceae—manifested predominantly positive interactions, underlining cooperative dynamics. In stark contrast, families like Nocardiaceae and *Alcaligenaceae* emerged as central to competitive dynamics.

The infected skin (Appendix Figure A5) showcased a remarkable interaction network, with 2282 negative and 1222 positive interactions. Herein, among the bacterial pathogen *P. salmonis* ASVs detected in infected salmon tissues, 18 positive and 48 negative interactions were observed, highlighting potential dysbiosis. This pattern, marked by the nearly equal negative and positive interactions among ASVs, underscores the intricacies of infection dynamics.

## 4. Discussion

We studied the skin microbiome of Atlantic salmon during *P. salmonis* infection. We found notable differences between healthy and infected fish using beta diversity analysis, specifically PCoA and PERMANOVA analysis (see Figure 3 and Figure 4). Progression of the ulcerative process due to *P. salmonis* is displayed in Figure 4. There is a stark contrast in the microbiome of healthy skin of salmon in control groups compared with that of ulcerated skin. Infections without ulcers fall in between these extremes in the representation. Yet, this pattern is not always consistent, as fish species and environmental conditions seem to impact it. For instance, Kashinskaya et al., 2021 [50], while studying Prussian carp and its ectoparasitic crustaceans, *Lernaea cyprinacea* and *Argulus foliaceus*, aimed to understand the microbiome alterations during coexisting ulcers and parasitic infestations. Their beta diversity analysis showed there were no significant differences in the microbiota of intact skin mucosa in affected fish, suggesting ulcers and ectoparasites did not alter the microbiome. However, they noted negative correlations between certain bacterial species and the number of ulcers. In our research, we found marked differences between the healthy and ulcerated skin of Atlantic salmon (with *p*-values of 0.00014 and 0.0016, respectively), indicating a substantial impact of the ulcerative process on skin microbiome diversity. 

Our examination of specific taxonomic groups by lesion type revealed surprising bacterial group involvement in the outbreak. Specifically, in advanced ulcerated skin stages, we found an overrepresentation of four bacterial families (refer to Figure 6): Vibrionaceae, Flavobacteriaceae, Piscirickettsiacea, and Pseudomonadaceae, all previously recognized as pathogens or opportunists in aquaculture. One of our most notable discoveries was the detection of the aquatic pathogen *A. wodanis* from the Vibrionaceae family (see Figure 8) in this Atlantic salmon ulcerative outbreak in Chile. Though *A. wodanis* is found in the core microbiome of the parasite *C. rogercresseyi* in Chile [51], it has not been a significant health issue for Chilean salmon farms. It predominantly affects salmon farms in Norway [52] and Canada [53]. *A. wodanis* is associated with ulcerative skin problems in farmed fish [50] and is considered a pathogen linked to winter ulcer disease in Atlantic salmon in the northern hemisphere.

The specific pathogenic role of *A. wodanis* in winter ulcer disease is still controversial. While some research points to *Moritella viscosa* as the primary pathogen behind winter ulcer outbreaks [47,48,49,50,51,52,53,54,55,56], the relationship between *A. wodanis* and *M. viscosa* is complex. For example, Karlsen et al., 2012 [57] explored the dynamics between these bacteria. They observed that the diminished presence of *A. wodanis* in healthy fish skin might be due to the propensity of *M. viscosa* to colonize ulcerated skin. Their findings imply that *M. viscosa* colonization might require pre-existing skin damage, often associated with secondary infections like “fin rot”. Additionally, the study raised the possibility that *A. wodanis* might produce toxins that could further compromise colonized surfaces, making them more prone to infections. Hjerde et al., 2015 [58] arrived at a similar conclusion in their research, emphasizing the multifaceted nature of winter ulcer disease, where “winter ulcer” typically refers to infections by *M. viscosa*. In a more recent study, Maharajan et al., 2021 [59] delved into the quorum sensing (QS) system of *A. wodanis*. They uncovered that *A. wodanis* produces virulence factors that might encourage inter- or intraspecies interactions, furthering competition and adaptation during the progression of winter ulcers.

We did not identify *A. wodanis* alongside *M. viscosa*, as *M. viscosa* is not a known issue in Chilean farms. However, we did identify *T. dicentrarchi* as a prominent microorganism in Atlantic salmon ulcers. This observation is depicted in Figure 8. Notably, *T. dicentrarchi* is classified within the Flavobacteriaceae family. This bacterium was first isolated from skin lesions of the European sea bass in Spain. Subsequent studies, like Olsen et al., 2017 [60], indicated that, in Norway, other *Tenacibaculum* spp. may overshadow *T. dicentrarchi*. This research also underscored the potentially higher pathogenicity of *T. dicentrarchi* in non-salmonid fish. *Tenacibaculum* has been recognized as a genus comprising various opportunistic pathogens, where factors such as dysbiosis, isolate variation, virulence, and host genetics can influence the colonization of affected fish skin [61,62]. *Tenacibaculum* spp. are often identified as part of various fish coinfection outbreaks. For instance, Apablaza et al., 2017 [63] identified *Tenacibaculum maritimum* as the causative agent of Tenacibaculosis in Chilean Atlantic salmon (*S. salar*), and this was associated with a *Pseudochattonella* spp. algal bloom. Similarly, Avendaño-Herrera., 2006 [64] noted that, during Tenacibaculosis outbreaks in *Platax orbicularis*, the presence of *T. maritimum* was commonly associated with co-occurrences of other pathogenic genera, specifically *Vibrio* spp., aligning with our findings. It has been suggested that certain *Tenacibaculum* species might be an integral part of specific fish species microbiomes, thereby creating a stable pathobiome that harmonizes with the host. Data from Wynne et al., 2020 [61] support this notion. In their study, *T. maritimum* was observed in the skin microbiome of healthy Atlantic salmon smolt as well as in those recovered from *Tenacibaculum* infection. This implies the remarkable adaptability of *T. maritimum* within its ecological niche. Moreover, several researchers have emphasized the presence of multiple species within the *Tenacibaculum* genus [65] in Atlantic salmon, alongside *Vibrio* spp. and *Aliivibrio* [55,66,67]. The diversity of phylogenetic relationships within *Tenacibaculum* sequences in our research further bolsters the idea of varied *Tenacibaculum* spp. cultures on Atlantic salmon skin. However, further supporting evidence is required before it can be categorically stated that *Tenacibaculum* spp. is a constant presence in the healthy microbiome. The fish skin microbiome is dynamic, with variations influenced by species, developmental stage [68], seasonal changes [68], water temperature [69], salinity [70], and geography [71]. As a result, the prevalence and diversity of *Tenacibaculum* spp. on the skin of healthy Atlantic salmon may exhibit considerable variation across different regions or farming areas, characterized by the unique environmental conditions of each.

Another bacterial genus of note in fish with skin ulcerations is *Pseudomonas*. *Pseudomonas* spp. are significant in the context of skin ulcerations in fish, often acting as opportunistic pathogens. Notable species like *Pseudomonas aeruginosa*, *Pseudomonas anguilliseptica*, and *Pseudomonas fluorescens* are primary agents in fish diseases [72]. Specifically, *P. fluorescens* and *Pseudomonas putida* have posed challenges in trout farming [73]. These species’ ability to form biofilms, a key factor in disease outbreaks, is often linked to their exopolysaccharide production and iron acquisition mechanisms. For instance, *P. aeruginosa*’s pyoverdine siderophore biosynthesis gene plays a critical role in iron uptake and biofilm development, suggesting potential interactions with other biofilm-forming bacteria like *Tenacibaculum* spp. [74,75]. However, the specifics of these interactions, particularly in Atlantic salmon ulcers, remain to be fully understood. Additionally, the study by Calquin et al., 2017 [76] on *P. salmonis* underscores the complexity of iron uptake mechanisms in these pathogens. 

*P. salmonis* was found to be the most prevalent among the ASVs in the infected and ulcerated tissues of Atlantic salmon (see Figure 7). This is consistent with its association with ulceration outbreaks in Chilean fish farms. Figure 9 illustrates the complexity of microbial diversity within ulcers and the intricate interactions among various bacterial families. Yet, the exact mechanisms driving these interactions between bacterial taxa are still unclear. To understand the microbial dynamics during ulcerative outbreaks fully, it is essential to explore how *P. salmonis* enters Atlantic salmon.

Previous research indicates that infections mainly occur in salmonids through the skin and gills, as suggested by Smith et al., 1999 [77] and Smith et al., 2004 [78]. They showed that *P. salmonis* can invade Coho salmon (*Oncorhynchus kisutch*) via the skin and mucous membranes, even without exhibiting visible damage. They hypothesized that *P. salmonis* may begin its infection process by adhering to microscopic lesions, which are common in cultured salmon [79]. This suggests that the skin is a potential primary entry point for this bacterium in salmonid fish.

Notably, there are limited reports of coinfections involving *P. salmonis* with other pathogens in aquaculture. In previous studies, coinfection interactions between *P. salmonis* and the parasite *C. rogercresseyi* have been explored in Chile [28,77,80,81]. However, bacterial coinfections have not been given the same attention, often due to the prevailing belief that outbreaks result from a single pathogenic agent. This oversight becomes even more significant given the diverse range of pathogens we have identified in the affected salmon tissues, suggesting possible bacterial coinfections.

In our investigation of *P. salmonis* presence in the gills of infected Atlantic salmon (*S. salar*), we detected ASVs corresponding to this pathogen. However, their presence was not statistically significant when compared with healthy gills. This suggests a minimal or non-impactful role of *P. salmonis* in altering gill microbiota under the conditions of our study. Interestingly, the bacterial diversity within the gill microbiome of the salmonids we examined demonstrated remarkable stability. This finding aligns with prior research, such as the study by Rosado et al., 2019 [82], who explored variations in the mucosal microbiome of seabass (*Dicentrarchus labrax*) during and post infectious outbreaks. Complementarily, recent work by Liao et al., 2023 [83] supports this observation. They found that the gill microbiome of marine medaka (*Oryzias melastigma*) remained unaltered when exposed to tetracycline and microplastics, suggesting resilience of the gill microbiome to certain environmental stressors. These comparisons draw attention to the varying responses of gill microbiomes across different species and environmental conditions, underscoring the complexity of host-microbe interactions in aquatic species.

While we have illuminated the microbial dynamics in advanced stages of the ulcerative processes in Atlantic salmon, we may have missed the subtleties of microbial interactions in early stages. This gap is further emphasized by our study’s limitation of using only two control sites, which may not fully represent the diversity and variability in different aquaculture environments. The choice of control sites, constrained by logistical and resource considerations, limits our ability to generalize our findings across varying geographical locations and production practices. Identifying multiple pathogens accentuates the importance of discerning their collective impact on infections, a dimension we did not deeply explore. Furthermore, the omission of potential external influencers of microbial communities, such as water temperature or farming practices, presents another layer of complexity not captured in our study. Our genomic methodology, pivotal in identifying specific microbial taxa, falls short in guaranteeing their viability, activity, or exhaustive detection. These limitations, while inherent to the scope of our research, highlight the need for future studies to incorporate a broader range of control sites and to consider external environmental factors, thereby enriching our understanding of microbial interactions in aquaculture.

## 5. Conclusions

In conclusion, our examination of the ulcerative outbreak in farmed Atlantic salmon linked to *P. salmonis* indicates the presence of multifaceted bacterial communities, with revelation of a core pathobiome that was previously unidentified. This underscores the potential of 16S rRNA analysis in discerning intricate bacterial interactions during infections. Comprehending the dynamics of pathogenic evolution in ulcers is pivotal for demystifying the infection mechanism of *P. salmonis* and tracing its colonization pathway within the host. This discovery raises pertinent questions about the specific roles of the detected infectious and opportunistic taxa—notably Vibrionaceae, Flavobacteriaceae, Piscirickettsiacea, and Pseudomonadaceae—in ulcerative onset. Thus, further research incorporating both in vitro and in vivo methodologies centered on identified species *T. maritimum*, *P. salmonis*, and *A. wodanis* is essential.

## Figures and Tables

**Figure 1 animals-14-00097-f001:**
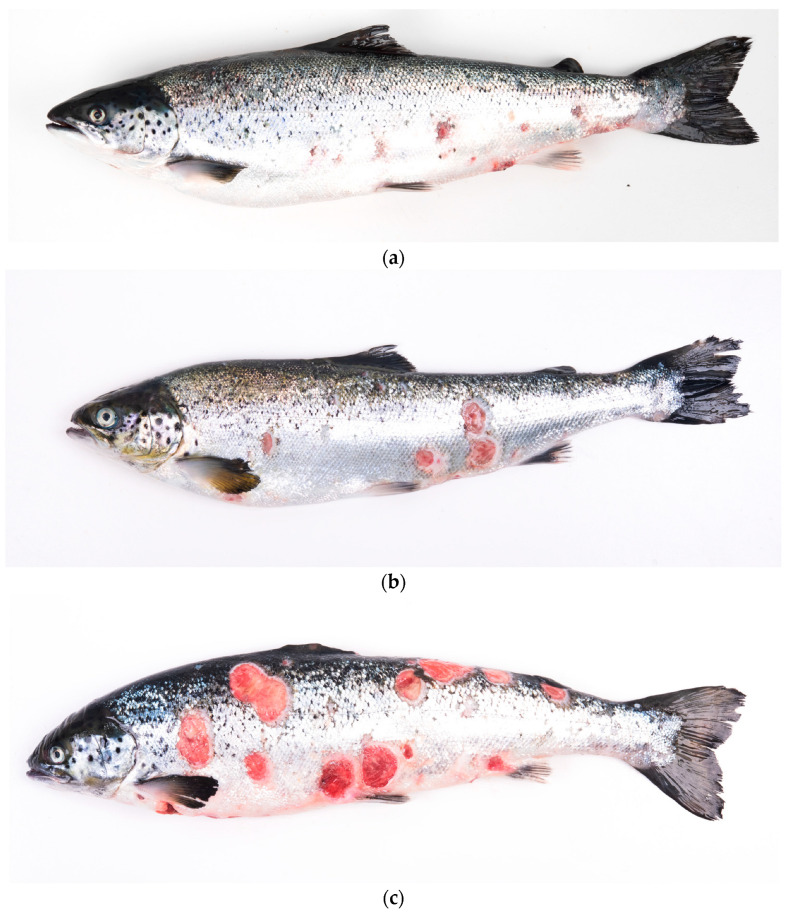
Atlantic salmon (*Salmo salar*) affected by clinical manifestations of salmonid rickettsial septicemia (SRS), characterized primarily by cutaneous signs (**a**–**c**). Affected fish exhibit raised scales; scaling; cutaneous hemorrhages; and the presence of multiple ulcers, some coalescent, varying in size, with distinct, white-bordered edges.

**Figure 2 animals-14-00097-f002:**
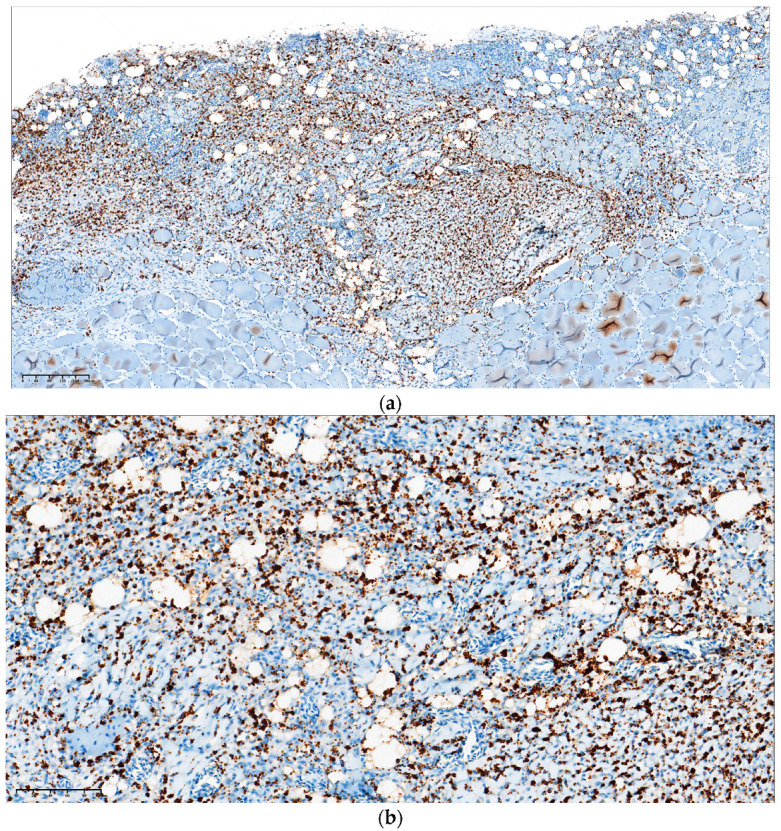
Atlantic salmon (*Salmo salar*) affected by cutaneous clinical presentation of salmonid rickettsial septicemia (SRS). Immunohistochemistry was performed on paraffin-embedded tissue using a monoclonal antibody specific to *P. salmonis*. A positive reaction is evident as brown staining (**a**,**b**) of coccoid structures (**a**,**b**), with invasion of the dermis, hypodermis, and adjacent musculature.

**Figure 3 animals-14-00097-f003:**
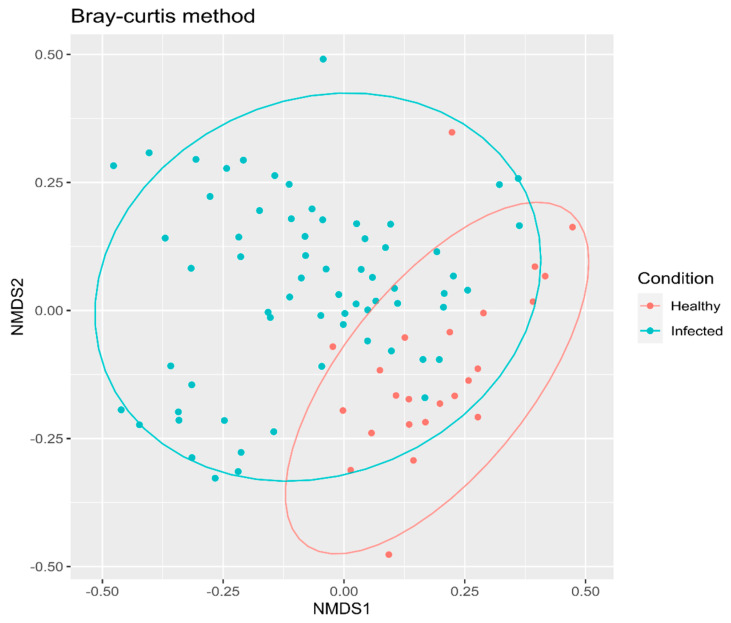
Non-metric Multidimensional Scaling (NMDS) ordination plots illustrating the distribution of amplicon sequence variants (ASVs) across the sampled salmon population, which was divided into two distinct groups: “healthy” Atlantic salmon (control group) and *P. salmonis*-infected Atlantic salmon. The NMDS analysis provides a visual representation of the compositional differences between the microbiota of these two groups.

**Figure 4 animals-14-00097-f004:**
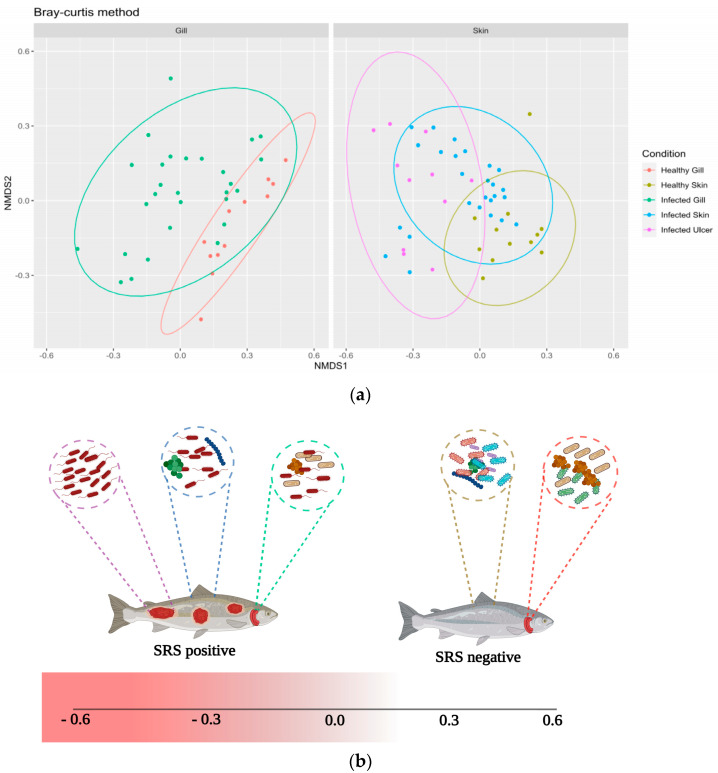
(**a**) Non-metric Multidimensional Scaling (NMDS) ordination plots representing the distribution of amplicon sequence variants (ASVs) within tissues of both “healthy” (control group) and *P. salmonis*-infected Atlantic salmon. Color-coded labels are used to differentiate the tissue types: salmon pink (healthy gill, control group), olive green (healthy skin, control group), green (infected gill, *P. salmonis*-infected group), blue (infected skin, *P. salmonis*-infected group), and purple (infected ulcer, *P. salmonis*-infected group). (**b**) Schematic representation of the ulcerative progression in Atlantic salmon (created with BioRender.com).

**Figure 5 animals-14-00097-f005:**
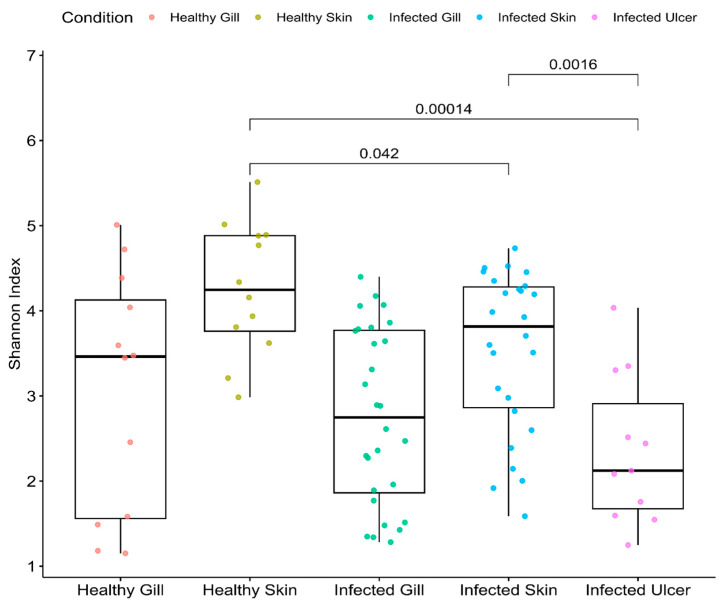
Microbial alpha diversity metrics by tissue (species richness and Shannon diversity), comparing the tissues of healthy Atlantic salmon (control group) and *P. salmonis*-infected Atlantic salmon: salmon pink (healthy gill, corresponding to the control group), olive green (healthy skin, corresponding to the control group), green (infected gill, corresponding to the *P. salmonis*-infected group), blue (infected skin, corresponding to the *P. salmonis*-infected group), and purple (infected ulcer, corresponding to the *P. salmonis*-infected group).

**Figure 6 animals-14-00097-f006:**
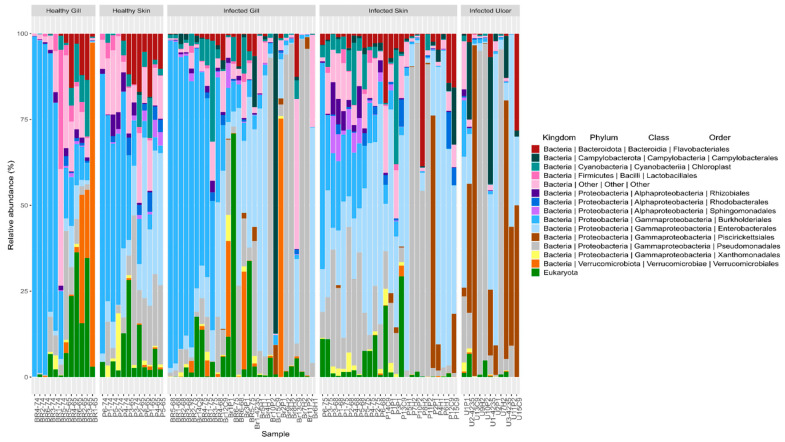
Mean relative abundances of amplicon sequence variants (ASVs) visualized at the bacterial phylum, class, and order levels in the tissues of healthy Atlantic salmon (healthy gill and healthy skin) and *P. salmonis*-infected Atlantic salmon (infected gill, infected skin, and infected ulcer).

**Figure 7 animals-14-00097-f007:**
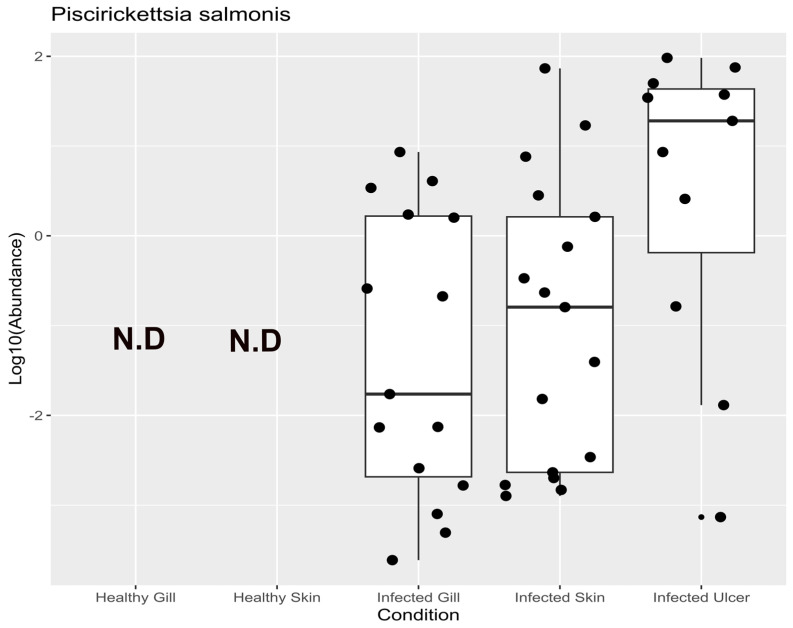
Mean relative abundances of amplicon sequence variants (ASVs) of *P. salmonis* in the tissues of healthy Atlantic salmon (healthy gill and healthy skin) and *P. salmonis*-infected Atlantic salmon (infected gill, infected skin, and infected ulcer).

**Figure 8 animals-14-00097-f008:**
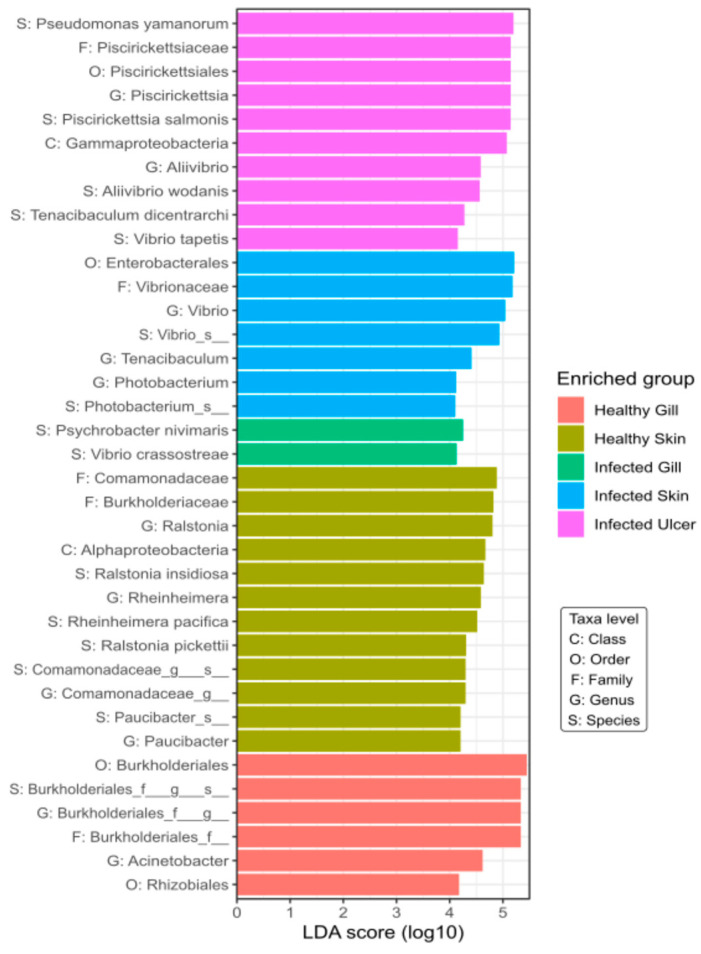
Linear discriminant analysis effect size (LEfSe) and linear discriminant analysis (LDA) characterization of microbiomes across tissues in “healthy” Atlantic salmon (healthy gill and healthy skin) and *P. salmonis*-infected Atlantic salmon (infected gill, infected skin, and infected ulcer). Tissue categories are color-coded: purple (infected ulcer, *P. salmonis*-infected group), blue (infected skin, *P. salmonis*-infected group), green (infected gill, *P. salmonis*-infected group), olive green (healthy skin, control group), and salmon pink (healthy gill, control group).

**Figure 9 animals-14-00097-f009:**
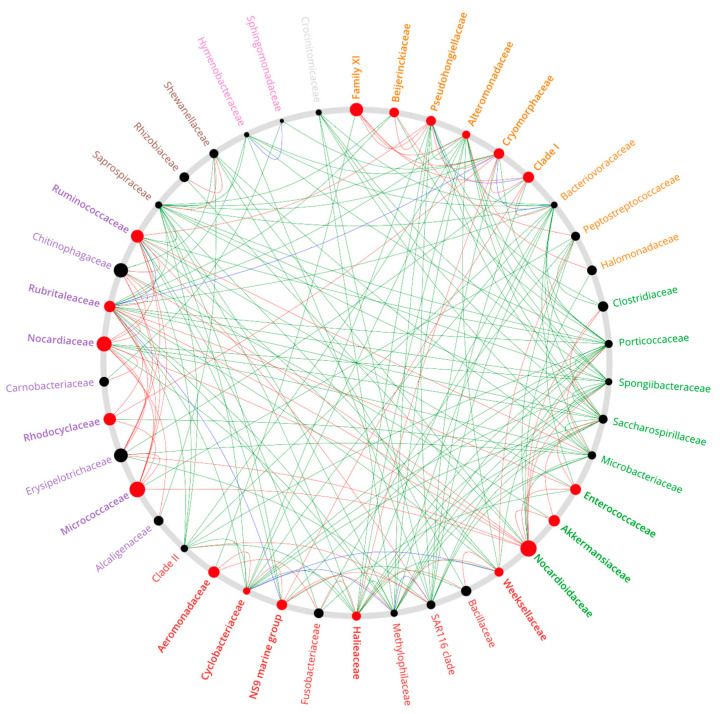
Co-occurrence network depicting the relationships among principal genera present in Atlantic salmon ulcer tissue compared to the healthy skin of the Atlantic salmon control group. Red nodes mark influential key genera; black nodes indicate less influential genera.

**Table 1 animals-14-00097-t001:** Atlantic salmon (*Salmo salar*) farms sampled in Los Lagos and Aysén Regions (Chile) during infectious salmonid rickettsial septicemia (SRS) outbreaks.

Farmed Sampled	Health Condition	Sampled Type	Region
C1	Infected with *P. salmonis*	Skin, gill, and ulcer	Los Lagos
C2	Infected with *P. salmonis*	Skin, gill, and ulcer	Los Lagos
C3	Infected with *P. salmonis*	Skin, gill, and ulcer	Aysén
C4	Infected with *P. salmonis*	Skin, gill, and ulcer	Los Lagos
C5	Infected with *P. salmonis*	Skin, gill, and ulcer	Los Lagos
C6	Infected with *P. salmonis*	Skin, gill, and ulcer	Los Lagos
C7	Infected with *P. salmonis*	Skin, gill, and ulcer	Aysén
C8	Control Farm	Skin and gill	Aysén
C9	Control Farm	Skin and gill	Los Lagos

## Data Availability

Raw data from this study is available at NCBI under the BioProject code PRJNA1044012.

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
