# Peer review of "Salmo salar Skin and Gill Microbiome during Piscirickettsia salmonis Infection"

_animals, 2023, doi:10.3390/ani14010097_

Round 1

Reviewer 1 Report

Comments and Suggestions for Authors

The article addresses important aspects towards understanding the role of complex microbiomes in disease progression of Piscirickettsia salmonis. It reports novel findings, is generally well written and is of interest and scientific merit. 

Title and abstract:

The subject is quite specific addressing changes in microbial diversity during P. salmonis outbreaks only. The title does not adequately reflect this specificity and suggests a more general review paper. I suggest amending the title to better reflect the subject matter.

Please complete second sentence of abstract - “promotes animal"…. what?

Materials and methods:

Please check the range on average weight of fish – is there a decimal place error?

Please indicate how many fish were sampled from infected farms?

What do the dot colours (red or black) signify in Figures 9 and A4 & A5? – please specify in M&M and in Figure legends.

Results:

The description for Figure 3 appears mixed up – red dots (healthy) are in the lower right quadrant, whilst blue green dots (infected) appear centre left.

My main scientific concern is whether two independent control sites are appropriate and sufficient for a comparison of this type when, as the authors mention in the discussion, diversity is likely to be impacted by geographical location, husbandry and environmental conditions?  What are the plots like when equal numbers of sites are compared - i.e. any 2 affected vs the 2 unaffected? Would more control sites add greater diversity and greater spread across red dots in the NMSD plots?  Was it not possible to sample healthy animals at the affected sites to add to the comparison?  The impact (and potential weaknesses)  of experimental design on the results should be explored and justified - at least in the discussion.

The description of data positioning for Figure 4 appears to contrast with data points on the plot. For example: there are more data points in the lower right than upper right for the healthy gill; the data are more dispersed but there are more data points in the left (14) than the right (12) for infected skin, there is also a subgroup in the lower left which has not been described; for infected ulcer the data points are left of centre on NMSD1 and more positive than negative on NMSD2.  The reviewer accepts that there are apparent groupings and differences between the grouping but unless I have misread these plots, please explain the discrepancies.

Figure 4b is not referenced in the manuscript text and I am not convinced the schematic adds to the manuscript.

Shannon index Figure 5, healthy skin appears olive green in the manuscript downloaded.

Figure 8 – again this reviewer struggled with colour description of the palate used. From top to bottom I would describe the colours as purple, blue, green, olive green, salmon pink. I think other readers may struggle in a similar way – suggest picking better differentiated colours.

The manuscript indicates no supplementary material yet there is some.

Comments on the Quality of English Language

The grammatical content of the manuscript is to a high standard, however there are areas of repetition in the justification for the work in the introduction that could be removed and the materials and methods could be more succinct.

Author Response

Dear Reviewer, attached file with suggestions and changes. Sincerely,

Reviewer 2 Report

Comments and Suggestions for Authors

This paper found significant differences in skin microbiome diversity between the healthy and ulcerated skin of Atlantic salmon. They found an overrepresentation of four bacterial families, including Vibrionaceae, Flavobacteriaceae, Piscirickettsiacea, and Pseudomonadaceae. This study is improving the understanding of trout ulcer disease caused by Piscirickettsia salmonis.

Minor points:

Abstract:

(1) The abstract is too concise.

2.1 Fish sampling

(1) Authors confirm the average weight data “average weight of 2869.34 g (±3160.42 g)”.

(2) How many skin samples in each farming site for 16S rRNA high-throughput sequencing?  

(3) The diseased fish should be detected the virus. Is there a possibility of coinfections in these diseased fish?

2.3. DNA Isolation, 16S rRNA Gene Amplification, and Sequencing

(1) Primers 515F/806R sequences and reference should be provided.

4. Discussion

(1) The discussion about Pseudomonas is tediously long and not necessary, e.g. “An illustrative example is P. aeruginosa, known to possess the pyoverdine siderophore biosynthesis gene. This gene is essential for iron acquisition and biofilm development in multi-bacterial environments. This aligns with findings from Levipan et al., 2019 [77], who found that T. dicentrarchi strains can form biofilms in vitro, hinting at a potential synergistic relationship between Pseudomonas spp. and Tenacibaculum spp.”

(2) In the discussion section, there is a lack of discussion of Flavobacteriaceae or Flavobacterium.

(3) In the discussion section, there is a lack of discussion of gill microbiome.

Others:

The Figures A1-A5 were not found in the text.

Author Response

(The authors gave the same response as above.)

Round 2

Reviewer 1 Report

Comments and Suggestions for Authors

Thank you for comprehensively addressing my concerns and suggestions. The limitations in the experimental design though not possible to rectify have at least been highlighted and appropriately discussed.